# Phenotypic screening using waveform analysis of synchronized calcium oscillations in primary cortical cultures

**Richi Sakaguchi**[1]*, **Saki Nakamura**[2], **Hiroyuki Iha**[3], **Masaki Tanaka**[1]*

**1** Department of Lead Discovery Research, New Drug Research Division, Otsuka Pharmaceutical Co., Ltd., Kagasuno, Tokushima, Japan, **2** Department of Research Management, New Drug Research Division, Otsuka Pharmaceutical Co., Ltd., Kagasuno, Tokushima, Japan, **3** Office of Bioinformatics, Department of Drug Discovery Strategy, New Drug Research Division, Otsuka Pharmaceutical Co., Ltd., Kagasuno, Tokushima, Japan

* Sakaguchi.Richi@otsuka.jp (RS); Tanaka.Masaki@otsuka.jp (MT)

**Data Availability Statement:** All files are available from the Dryad Digital Repository at doi:10.5061/dryad.d2547d853 (https://doi.org/10.5061/dryad.d2547d853).

## Abstract

At present, *in vitro* phenotypic screening methods are widely used for drug discovery. In the field of epilepsy research, measurements of neuronal activities have been utilized for predicting efficacy of anti-epileptic drugs. Fluorescence measurements of calcium oscillations in neurons are commonly used for measurement of neuronal activities, and some anti-epileptic drugs have been evaluated using this assay technique. However, changes in waveforms were not quantified in previous reports. Here, we have developed a high-throughput screening system containing a new analysis method for quantifying waveforms, and our method has successfully enabled simultaneous measurement of calcium oscillations in a 96-well plate. Features of waveforms were extracted automatically and allowed the characterization of some anti-epileptic drugs using principal component analysis. Moreover, we have shown that trajectories in accordance with the concentrations of compounds in principal component analysis plots were unique to the mechanism of anti-epileptic drugs. We believe that an approach that focuses on the features of calcium oscillations will lead to better understanding of the characteristics of existing anti-epileptic drugs and allow to predict the mechanism of action of novel drug candidates.

## Introduction

The central nervous system (CNS) has critical roles in homeostatic regulation, then its failure causes serious effects in human bodies [1, 2]. Consequently, CNS is one of the most active therapeutic area of research and many drugs have been developed in recent decades although the failure rate of CNS drugs is higher than other areas [3]. Historically, *in vivo* screening was used in the early stage of drug discovery, while *in vitro* functional screening systems have recently been developed and have contributed to increasing throughput. For example, neuronal activities including membrane potential and postsynaptic currents were also used to develop compounds that act on ion channels, using a plate reader system and an automated patch clamp

**Funding:** Otsuka Pharmaceutical Co., Ltd. provided support in the form of salaries for RS, SN, HI, and MT. The specific roles of these authors are articulated in the 'author contributions' section.

system [4]. In addition, phenotypic differences in morphological parameters and protein aggregation detected by a high-content imaging system were applied in discovering amyotrophic lateral sclerosis (ALS) drug candidates [5]. Phenotypic screening systems are useful to discover novel drug targets or mechanism of action (MoA) because these systems do not rely on known targets or therapeutic hypotheses in diseases [6]. In contrast, the challenge in these approaches is to clarify why the identified candidates are effective. In pharmaceutical companies, identifications of the mechanism and target of candidates tend to be required for internal project prioritization, and the unknown mechanism or target is considered a major risk factor for clinical development and regulatory approval [6]. However, few studies have been reported on cell-based assays focused on CNS for deconvolution of MoA.

Epilepsy is one of the most common neurological diseases. The World Health Organization (WHO) reported about 50 million people suffer from epileptic seizures all over the world and at least 30% of patients have drug-resistant epilepsies [7]. Anti-epileptic drugs (AEDs) which have novel characteristics may contribute to therapy for drug-resistant patients [8, 9]. In addition, long-term use of AEDs may lead to problems such as side effects, tolerance, drug interactions, and so on [10]. For these reasons, there is a need to develop AEDs with new characteristics. To conduct pharmacological tests of anti-seizure compounds, multi-electrode array (MEA) systems are often used because epileptic seizures are accompanied by increasing synchronized burst firings [11, 12]. In particular, toxicity studies focus on a few candidates, thus the number of samples tends to be small. In addition, the analysis targets mainly convulsant-inducing effects. For this purpose, MEA is a very powerful analytical instrument because it has been reported to detect synchronized burst firing for *in vitro* seizure liability [11, 12]. On the other hand, in the early stages of drug discovery research, it is common to screen more compounds than toxicity studies, and the use of 96- and 384-well multi-well plates makes it easy to compare lots of compounds. Although 96-well MEA plates are practically used, just 3–4 electrodes are placed in each well and, in principle, signals can be obtained from neurons attached to electrodes. Therefore, we decided to focus on calcium oscillations as a possible solution to these issues. Synchronized calcium oscillations were observed in cultured cortical neurons [13, 14] and calcium oscillations are driven by bursts of action potentials in neuronal cultures [15]. A previous report showed that calcium oscillations had the potential to be incorporated into phenotypic screening methods for characterizing some compounds including AEDs [16]. However, lack of quantitation complicated the interpretation of the experimental results. As an improvement, we have developed a new high-throughput screening system that uses the signals of calcium oscillations in cultured cortical neurons.

In the present study, we achieved simultaneous measurement of calcium oscillations in 96-well plates, using an FDSS/μCell to optimize high-throughput screening. Moreover, we quantified the number and the shape of calcium oscillations as automatically extracted features and classified commonly used AEDs based on MoAs.

## Materials and methods

### Ethic statement

All experiments involving animals were performed in accordance with "Guidelines for Animal Care and Use in Otsuka Pharmaceutical Co, Ltd." And, all experiments were approved by "Pharmaceutical Business Division, Otsuka Pharmaceuticals Co., Ltd. Institutional Animal Care and Use Committee".

## Compounds

Perampanel and lacosamide were synthesized at Otsuka Pharmaceutical Co., Ltd. Levetirace-tam (#L0234) was purchased from Tokyo Chemical Industry Co., Ltd.; brivaracetam (#B677645) from Toronto Research Chemicals; diazepam (#D0899) from SIGMA; clonazepam (#038–17231) from FUJIFILM Wako Pure Chemical Corporation; lamotrigine (#L0349) from LKT Laboratories. All compounds were dissolved in dimethyl sulfoxide (DMSO) (#043–07216, FUJIFILM Wako Pure Chemical Corporation) to make up stock solutions. And we determined concentrations of stock solution so that the highest concentration of DMSO in buffer is 0.1%.

## Reagents

The composition of Tyrode's buffer was 140 mM NaCl, 4 mM KCl, 1.8 mM $CaCl_2$, 5 mM HEPES, 0.33 mM $NaH_2PO_4$, 0.1 mM $MgCl_2$ and 5.5 mM glucose. HEPES was purchased from Dōjindo Laboratories; NaCl, KCl, $CaCl_2$, $NaH_2PO_4$, $MgCl_2$ and glucose from FUJIFILM Wako Pure Chemical Corporation. B-27™ Plus Neuronal Culture System (#A3653401, Gibco) containing penicillin-streptomycin solution (#P4333, Sigma-Aldrich) was used to culture primary cells, and Neurobasal medium (#21103–049, Gibco) containing B27 supplement (#17504044, Gibco), GlutaMAX (#35050061, Gibco) and penicillin-streptomycin solution was used to harvest the cells.

## Primary neuronal culture

Rat primary cortical neuronal cultures were established from rat embryos (Crl: CD (SD)) at embryonic day 18. The cortical neurons were dissected using the Papain Dissociation System (Worthington Biochemical Corp). In brief, brain cortex was collected in ice-cold HBSS and dissociated in 0.25% papain/Dnase I in PBS for 30 minutes in the incubator, Neurobasal medium was added to 10 mL and pipetted 10 times. The supernatant was collected, filtrated using a cell strainer (#352350, Falcon), and centrifuged (1,000 rpm, 5 min). The pellet was suspended in Ovomucoid protease inhibitor/HBSS, centrifuged (800 rpm, 5 min) and resuspended in the medium for harvesting.

The dissociated cells were seeded onto 60 wells (all wells except the perimeter wells on 96-well plates (#3842, Corning)) at a density of 75,000 cells/well. Cell cultures were maintained inside a 5% $CO_2$ incubator at 37°C and half of the medium was replaced once every 3–4 days. For all experiments, cultures were used at 14 days *in vitro* (DIV 14).

## Calcium imaging

A vial of Cal-520 (#21130, AAT-Bioquest) was resuspended to 2 μM in Tyrode's buffer (0.1 mM $Mg^{2+}$) containing 0.01% Pluronic F-127 (#P3000MP, Invitrogen). The cultures were washed three times in Tyrode's buffer, and incubated in 80 μL of Cal-520 per well for 1 hour at 37°C. Culture plates were set to an assay stage and stabilized for more than 10 minutes before measurement.

For calcium imaging, an FDSS/μCell (Hamamatsu Photonics) kinetic plate reader was used. After incubation, 20 μL of compounds dissolved in Tyrode's buffer were dispensed using the 96 dispenser head of the FDSS/μCell and, 5 or more minutes after dispensing, calcium signals were measured for 5 minutes at 27 Hz using the following settings: exposure time 36.5 ms, excitation wavelength 480 nm, emission wavelength 540 nm, temperature controlled at 37°C.

### Data analysis

Fluorescence intensity data were extracted using FDSS/μCell software. Data analysis and visualization were processed using the "Wave Finder" in Spotfire (Data Visualization & Analytics Software—TIBCO Spotfire, http://spotfire.tibco.com/) and custom code written in R for PCA. PCA features were selected based on correlation of features and visual inspection of each waveform. Statistical analysis of each feature was performed and visualized with GraphPad Prism 7 (GraphPad Software).

## Results

### Synchronized calcium oscillations were detected simultaneously under 60 conditions

In previous reports, spontaneous intracellular calcium oscillations of cultured neurons in 96-well plates have been observed [16, 17]. To reproduce *in vitro* calcium oscillations in our hands, we determined the appropriate experimental conditions in our laboratory setting. We cultured rat primary cortical neurons in 96-well plates and observed calcium oscillations using the fluorescent calcium indicator Cal-520 dissolved in Tyrode's buffer (Fig 1A). To achieve measurements of reliable and homogenous calcium oscillations, we measured at DIV 14 in reference to previous reports [16–19]. We have succeeded in observing calcium oscillations synchronized in each well and measuring them in a maximum of 60 wells simultaneously using the FDSS/μCell (S1 Movie). Furthermore, we optimized the concentration of magnesium in Tyrode's buffer to evaluate many peaks. 0.1 mM $Mg^{2+}$ condition significantly increased the number of peaks than 1 mM $Mg^{2+}$ condition (Fig 1B and 1C). Additionally, coefficient of variation (CV) of the number of peaks for 0.1 mM $Mg^{2+}$ condition tended to be lower than for 1 mM $Mg^{2+}$ condition. We therefore decided to measure calcium oscillations under these conditions for all subsequent pharmacological experiments.

### Features extracted from calcium oscillations can characterize anti-epileptic drugs

It was reported that calcium oscillations were affected by compounds, which modulate the activity of ion channels [16, 17]. In particular, the frequency and shape of calcium oscillations varied depending on the compound. We considered a phenotypic cell-based assay for characterizing AEDs using calcium oscillations. To evaluate the shapes of calcium oscillations, we extracted six features of the waves using Wave Finder software (Fig 1D and 1E). The features are mean peak height (Mean Peak Height), CV of peak height (CV of Peak Height), the ratio of peak height to peak width (Mean Peak Height / Peak Width), the peak width from the time of the maximum value to the time when the signal drops to half of this value (Mean Peak Width 50%), total area under the curve (AUC), and the number of peaks (Peak Number) in each well. These six features were extracted for all detected peaks. We used positive allosteric modulators (PAM) of GABAergic inhibitory activities, diazepam (10 μM) and clonazepam (10 μM), inhibitors of excitatory inputs via voltage-gated sodium channels, carbamazepine (30 μM), lamotrigine (10 μM) and lacosamide (100 μM), and the widely used AEDs levetiracetam (100 μM), brivaracetam (100 μM) and perampanel (10 μM) to characterize their features (Fig 2A). The concentration of each compound was selected according to the therapeutic range of the concentration in plasma [20]. Each compound was applied to 6 wells in a plate by the FDSS/μCell auto-dispenser, incubated for 5 minutes and the signals of calcium oscillations recorded for 5 minutes. Compared to wells treated with DMSO (0.1%), the waveforms of calcium oscillations in wells to which some compounds were treated varied (Fig 2B). We

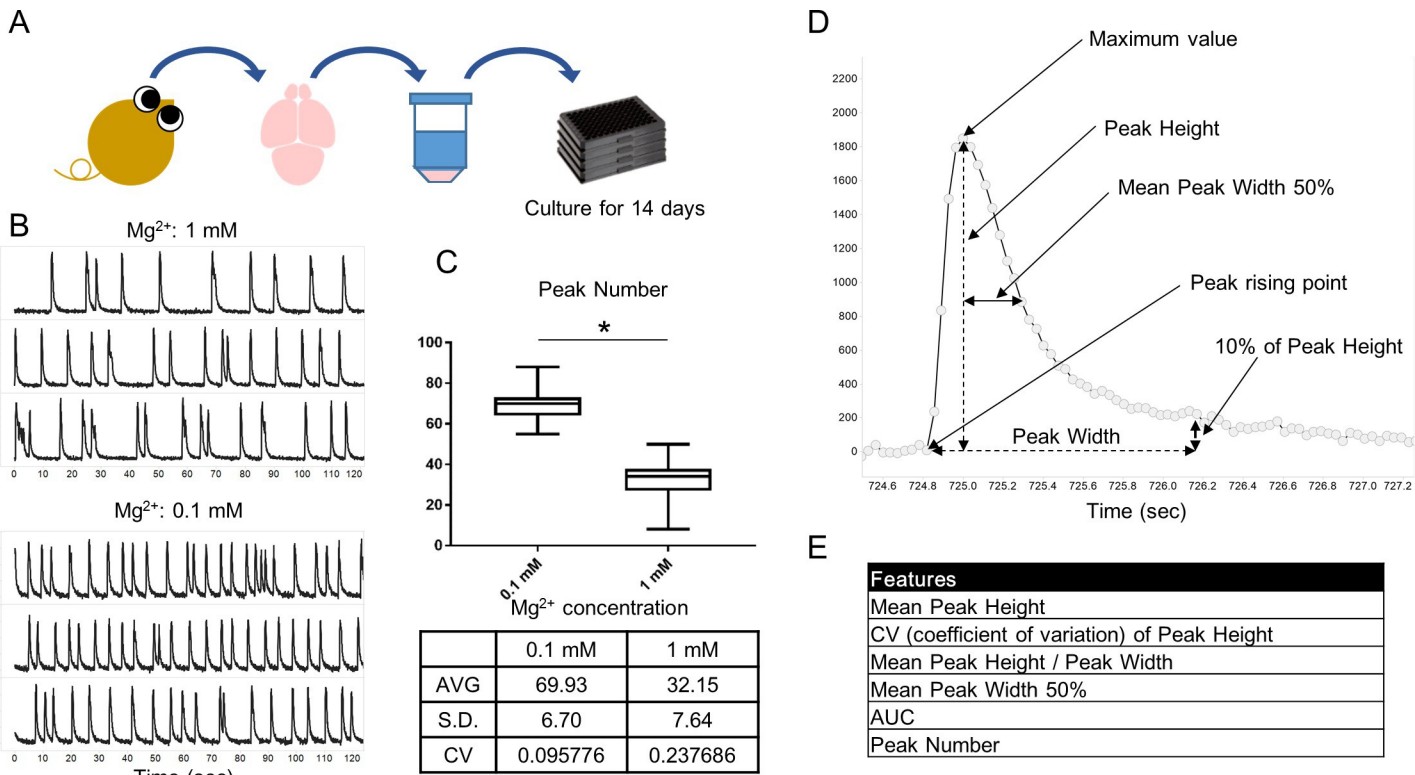

**Fig 1. The schema for simultaneous measurement of calcium oscillations in 96-well plates and methods for extracting wave features.** (A) Schematic of our method for measurement of calcium oscillations. Rat primary cortical neurons were harvested and cultured to DIV 14. Cultures were loaded with calcium indicator Cal-520 and measured in 60 wells (all except the perimeter wells on 96-well plates). (B) Representative calcium oscillations in 1 mM or 0.1 mM magnesium in Tyrode's buffer. (C) Descriptive statistics and box plots of peak numbers in each condition. The horizontal line in each box represented the median location, the box represented the interquartile range, and the whiskers showed the minimal and maximal values. $p^* < 0.05$ (unpaired t-test, n = 60 per condition). AVG, average; S.D., standard deviation; CV, coefficient of variation. (D) A representative calcium oscillation and definitions of wave features. Calcium oscillations were visualized and each feature extracted using Wave Finder in Spotfire. Features are defined as follows. Mean Peak Height is the difference between the maximum value of the peak and the height of the point at which the peak starts rising. Mean Peak Height / Peak Width was the ratio of the peak height to the peak width. Peak Width was the time from the point at which the peak starts rising to the time when the signal decreased to 10% of the peak height. Mean Peak Width 50% was the time from the maximum value to the time when the signal decreased to the half of this value. AUC was calculated using the trapezoidal rule. (E) The list of features extracted from calcium oscillations.

observed that each compound had the unique pattern of waveforms and calculated features from all detected peaks by Wave Finder software. Therefore, we confirmed that features were changed substantially (Fig 2C). For example, mean peak heights significantly decreased in wells treated with perampanel, diazepam, clonazepam or carbamazepine, but increased in wells to which brivaracetam, lamotrigine, and lacosamide were applied. In addition, peak numbers increased on treatment with diazepam, clonazepam, and lacosamide. Conversely, no features were changed on treatment with levetiracetam. The results were consistent with the previous report [16]. We could thus characterize these AEDs by the features of the peaks of their calcium oscillations.

To characterize these drugs robustly, we next compared the features of calcium oscillations caused by compounds using PCA. PCA is a useful method for evaluating multiple features simultaneously. The PCA plots showed that the clusters of some compounds were clearly seg-regated in three independent experiments (Figs 3A and S1A and S1B). The relative contributions of the principal components and contribution of the different features to the PC1 and PC2 are depicted (Figs 3B and 3C and S1). Two components have an eigenvalue more than 1. Together, two components explained more than 80% of the variability in each experiment.

A

| Compound | Main mechanism |
|----------|----------------|
| Levetiracetam | SV2A modulator |
| Brivaracetam | SV2A modulator |
| Perampanel | AMPA receptor antagonist |
| Diazepam | GABA$_A$ receptor agonist |
| Clonazepam | GABA$_A$ receptor agonist |
| Carbamazepine | Voltage-gated sodium channel inhibitor |
| Lamotrigine | Voltage-gated sodium channel inhibitor |
| Lacosamide | Voltage-gated sodium channel inhibitor |

B

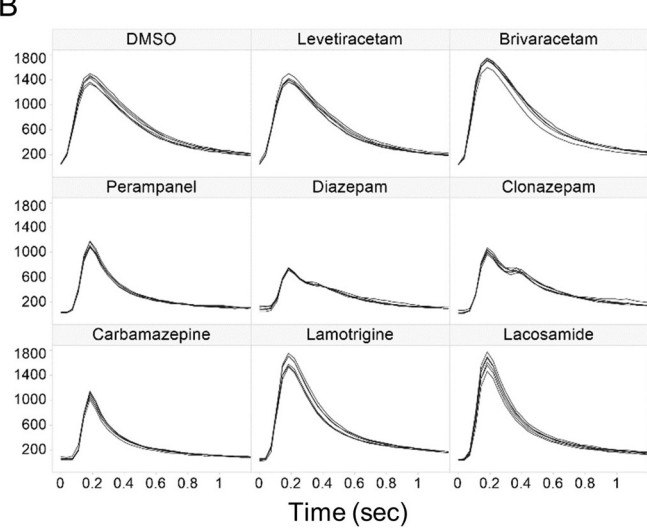

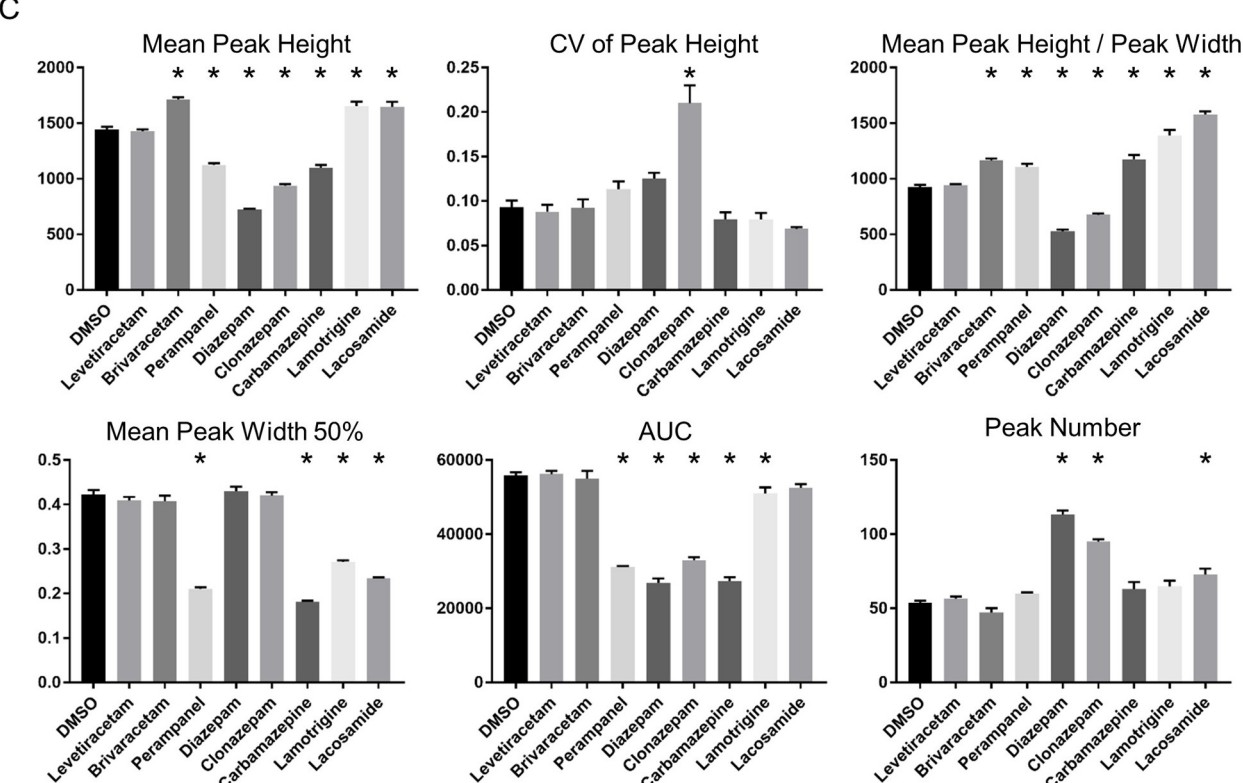

**Fig 2. The changes of wave features in response to treatment with AEDs.** (A) Summary of the AEDs used in our calcium oscillations assay. (B) Overlaid traces of peaks of calcium oscillations exposed to DMSO and AEDs. A trace represents an normalized trace in a well. X-axis represents peak heights in arbitrary units (a.u.), while y-axis represents time in seconds. (C) Bar plots of features extracted from wells to which AEDs were applied. Features were calculated from 6 wells per AED containing DMSO. Data are means ± SEM (n = 6 wells). p*<0.05 vs. DMSO (n = 40–123 peaks per well, one-way ANOVA with Dunnett post-hoc test).

The CV of the Peak Height and Peak Number made a positive contribution and the Mean Peak Height and AUC contributed negatively to PC1. We therefore suggest that these features were critical for differentiating PAMs of GABA$_A$ receptors. In addition, Mean Peak Width

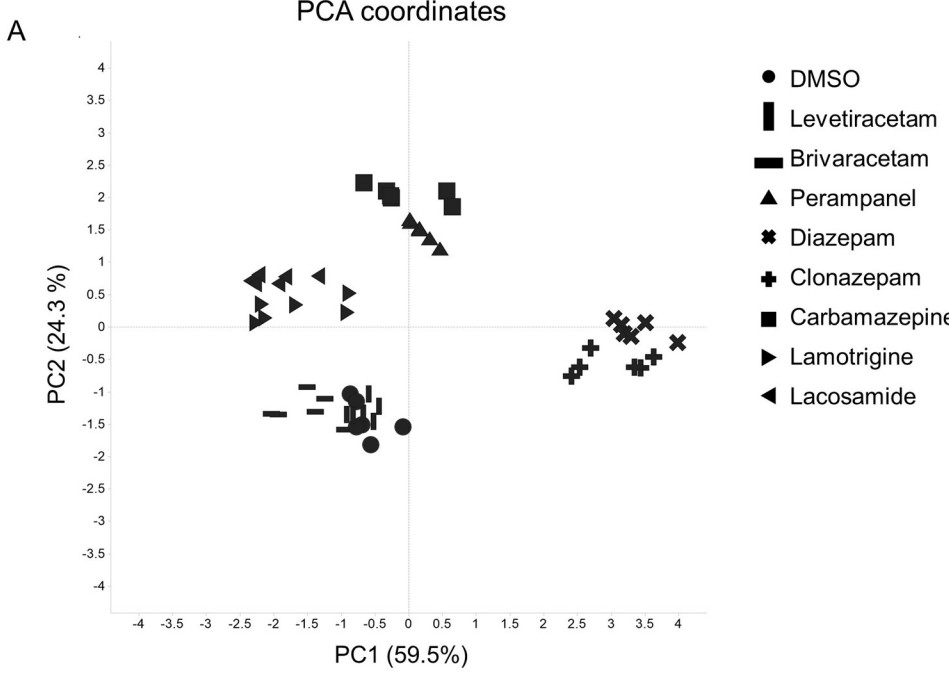

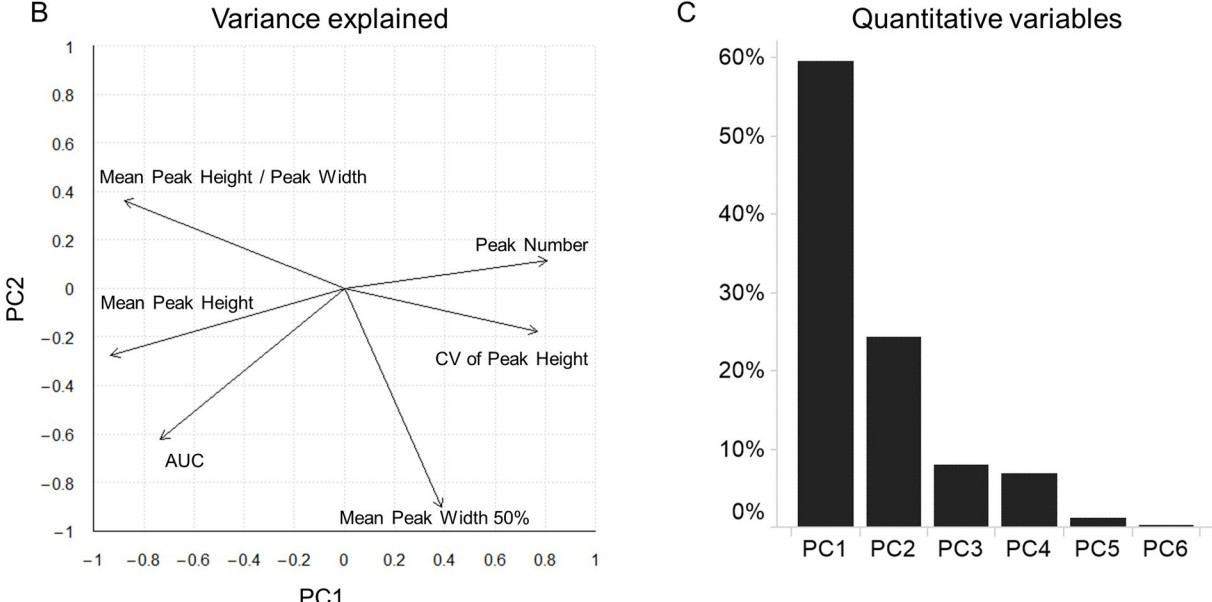

**Fig 3. PCA coordinates and contribution plots in response to treatment with AEDs.** (A) PCA of wave features. The individual wells are plotted. Different AEDs are represented by different shapes. (B) Plot depicting the contribution of features to PC1 and PC2 (arrows). (C) Bar plots displaying the percentage of variance explained by each component.

50% made a strong contribution to PC2 and this feature had a large effect on segregating inhibitors of excitatory inputs. On the other hand, levetiracetam and brivaracetam were not segregated from DMSO. This is reasonable because their features were not substantially changed, compared with DMSO. Thus, AEDs were characterized by the features of calcium oscillations.

### The concentration of a compound is critical in determining the features of calcium oscillations

Next, in order to obtain the information of waveforms changed in a dose-dependent manner, we examined the concentration dependence of each compound. Levetiracetam and brivaracetam were excluded from this experiment because these compounds had negligible effects on the features of calcium oscillations in previous assays. We applied 6 compounds at 3 concentrations to 3 wells per condition, including DMSO at a single concentration. We extracted features from all detected peaks using the previous methodology (Fig 4). On PCA plots, plots at low concentrations were located near the DMSO plot (Figs 5A and S2). The contribution of the different features to the PC1 and PC2 and the contribution ratio of principal components are depicted (Figs 5B and 5C and S2). As the concentration increased, plots were separated from the DMSO plot. As we expected, specific compounds move along different trajectories. Moreover, the clusters of diazepam and clonazepam, targeting the GABA receptor, moved in the same directions as concentrations increased. A similar tendency was seen with lamotrigine and lacosamide because their targets are voltage-gated sodium channels. Additionally, the trajectory of perampanel moved in a similar direction to that of sodium channel blockers. The target of perampanel is the AMPA receptor, which mediates excitatory synaptic transmission. Both sodium channel blockers and AMPA receptor antagonists inhibit neurohyperexcitability. These findings therefore suggested that our method reflected their MoA to some extent. Our method allowed us to characterize AEDs based on their MoAs.

## Discussion

It is known that intracellular calcium oscillations result from bursts of action potentials [15]. In previous reports, calcium oscillations of cortical cultures were used as functional assays for drug screening [16, 17, 19], and their changes were observed when several AEDs, such as GABA receptor agonists and inhibitors of excitatory inputs, were applied [16]. In this report, we quantified differences between AEDs using waveforms of calcium oscillations and classified their mechanisms based on their effects on excitatory inputs and inhibitory inputs.

Our method used features extracted by automatically quantifying the shape and number of calcium oscillations. To process these data, we conducted PCA analysis. It has great impacts for simplifying the complexity in high dimensional data without losing tendencies and patterns of the data. PCA analysis helps to evaluate several compounds visually. To classify compounds robustly in samples which have the variability of features and drug responses among sampling dates (Figs 2C and 4), the six features we used were indispensable for characterizing some AEDs even though some features had similar directions and length in plots of variance explained in some experiments (Figs 3 and 5 and S1 and S2). Also, we noted that it was important to determine optimal concentrations for each compound because PCA plots were not classified clearly at low concentration (Figs 5 and S2) and calcium oscillations disappeared when some compounds were treated at higher concentration than we used in this report (data not shown). In addition, our assay was able to characterize AEDs based on their MoAs because PCA plots of diazepam and clonazepam were located near certain coordinates and moved along similar trajectories in a concentration-dependent manner. Lamotrigine and lacosamide, known to be voltage-gated sodium channel blockers, also showed similar trajectories. In contrast, carbamazepine was segregated from them. The difference was possibly because carbamazepine has effects both on $GABA_A$ receptors as a positive allosteric modulator and on GABA release [21, 22]; while few studies reported that lamotrigine and lacosamide have effects on GABA currents with acute treatment. In addition, we could not observe substantial changes in calcium oscillations due to levetiracetam or brivaracetam. In some *in vitro* epileptic models,

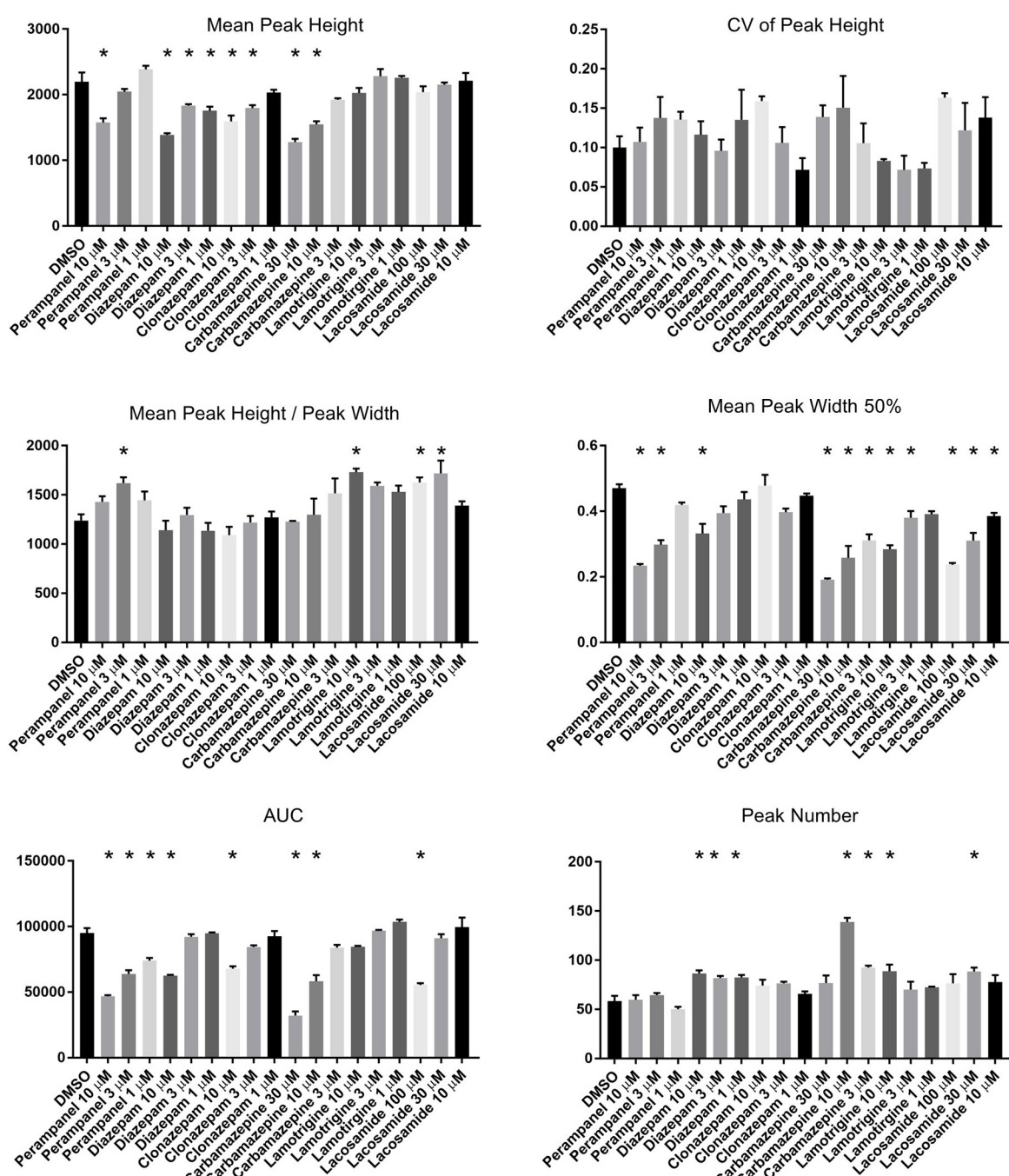

**Fig 4. The changes of wave features in response to treatment with AEDs at three concentrations.** Bar plots of features extracted from wells to which AEDs were treated. Features were calculated and averaged from 3 wells per AED under each condition, also containing DMSO. Data are means ± SEM (n = 3 wells). p*<0.05 vs. DMSO (n = 47–145 peaks per well, one-way ANOVA with Dunnett post-hoc test).

it has been shown that levetiracetam does not influence synaptic current or calcium oscillations on acute exposure at 100 μM [16, 23, 24], findings consistent with our observations. Brivaracetam also caused little change in calcium oscillations because the target of brivaracetam is the same as levetiracetam [25, 26].

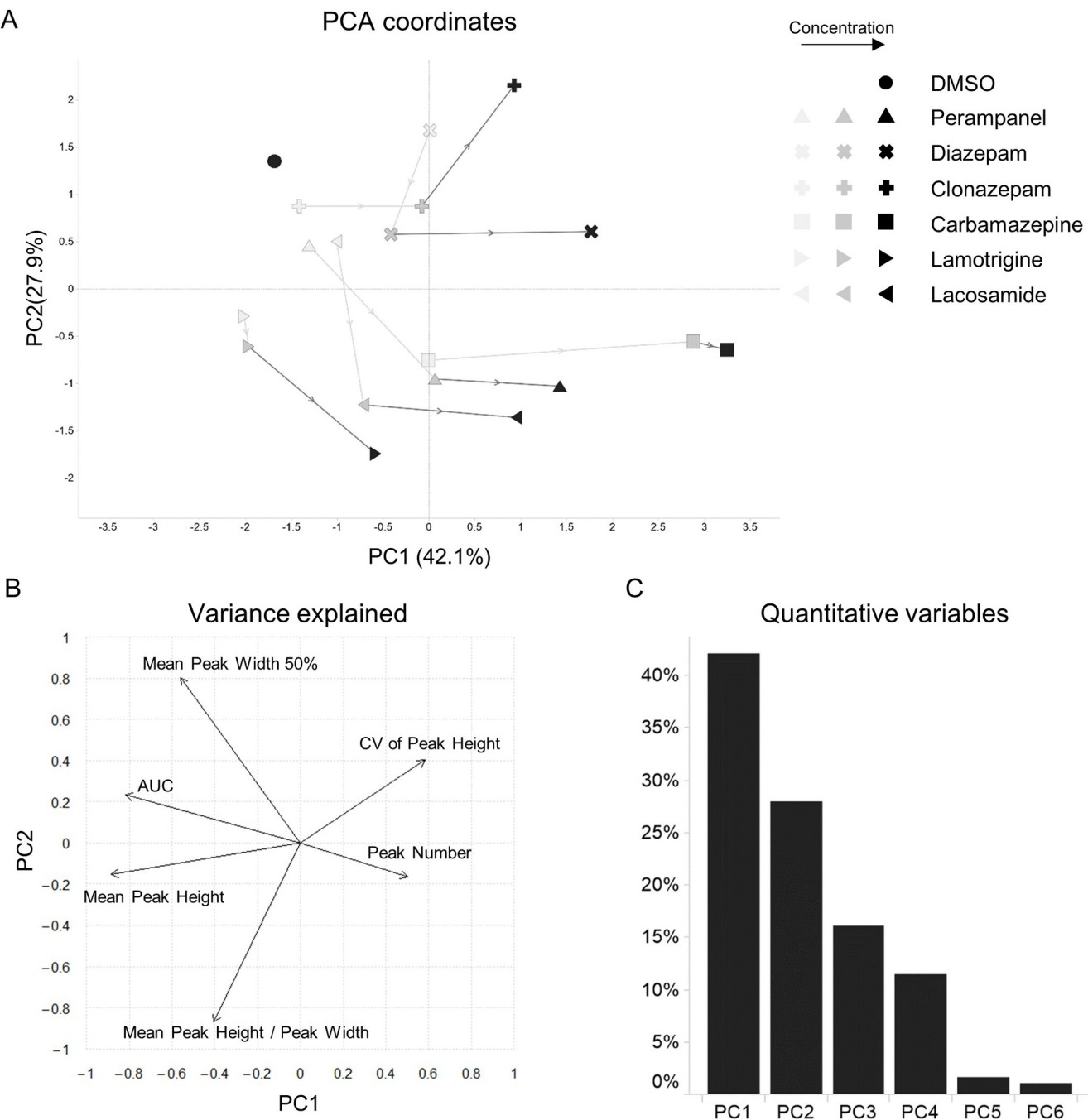

**Fig 5. Trajectories of PCA plots in accordance with concentrations of AEDs.** (A) PCA of wave features. The centers of wells are depicted under the same conditions. Different AEDs display different shapes. Light colors depict low concentrations, whereas dark colors represent high concentrations. Plots of the same AEDs are connected by lines in order of the concentration used. (B) Plot depicting the contribution of features to PC1 and PC2 (arrows). (C) Bar plots displaying the specific proportion of total variance explained by each PC.

Our waveform analysis can also be combined with other analyses. For example, simultaneous measurements using MEA can reveal the relationship between calcium dynamics and electrical activities. It can also be combined with indicators of other ions and transmitters to reveal novel aspects of neurophysiology. Thus, we propose an experimental flow that narrows

down drug candidates from hundreds to thousands by waveform analysis, followed by detailed MoA analysis using MEA or microscopy system, which has higher sensitivity and gives high temporal or spatial resolution.

Furthermore, our method might become a useful tool for evaluating drugs like antidepressants, which modulate neuronal activities, as it has been reported that the pattern of calcium oscillations was modified by addition of MK801, an inhibitor of NMDA receptors [16].

We noted that our waveform analysis of calcium oscillations may have a limited resolution compared with MEA because the signals of calcium are not identical with field potentials. Instead of high-resolution, measurement of calcium signals achieves greater throughput at a low cost because it doesn't requires costly specialized microplates and instruments. Also, since primary cultures are not identical among sampling dates, some variabilities of changes in features due to differences in drug responses from experiment to experiment must be considered (Figs 2C and 4). Therefore, we confirmed the segregation of compounds in several independent experiments (Figs 3 and 5 and S1 and S2). Besides, due to loading chemical dyes, it is difficult for our method to conduct repeated measurements unlike MEA analysis. Using genetically encoded calcium indicators like GCaMP6f will help at this point [27]. In addition to experimental limitations, features used in our analysis may be insufficient for describing the other phenomena than here like multiple bursts within seconds. It will be important to increase measurable features in order to express various waveforms and to characterize more compounds.

In conclusion, the present study showed changes in the number and shape of calcium oscillations, representing characteristics of various AEDs. In our method, cultured neurons can be replaced by other primary cells like cardiomyocytes, cells from model mice, or iPS cell-derived cells to reproduce pathologies [27, 28]. The measurement and analysis methods we describe here will be useful for characterizing and predicting MoAs of novel and existing CNS drugs, including AEDs.

## Supporting information

**S1 Fig. PCA coordinates and contribution plots of the other two replicates, related to Fig 3.** Independent experiments were conducted more than three times. A and B represent the datasets of independent experiments.
(TIF)

**S2 Fig. PCA coordinates and contribution plots of the other two replicates, related to Fig 5.** Independent experiments were conducted more than three times. A and B represent the datasets of independent experiments.
(TIF)

**S1 Movie. Simultaneous measurement of calcium oscillations in 60 wells.**
(ZIP)

## Acknowledgments

We deeply appreciate Mr. Yusuke Kakumoto for advising on the statistical analyses, Mr. Hiroki Fukuta, Dr. Takahito Inoue, Dr. Tetsutaro Sumiyoshi and Dr. Yosuke Nao for preparing primary cells dissociated from neonatal rats.

## Author Contributions

**Conceptualization:** Masaki Tanaka.

**Data curation:** Richi Sakaguchi, Saki Nakamura.

**Formal analysis:** Richi Sakaguchi.

**Investigation:** Richi Sakaguchi, Saki Nakamura, Hiroyuki Iha, Masaki Tanaka.

**Methodology:** Richi Sakaguchi, Saki Nakamura, Hiroyuki Iha, Masaki Tanaka.

**Project administration:** Masaki Tanaka.

**Software:** Richi Sakaguchi, Hiroyuki Iha.

**Supervision:** Masaki Tanaka.

**Validation:** Richi Sakaguchi, Saki Nakamura, Hiroyuki Iha.

**Visualization:** Richi Sakaguchi.

**Writing – original draft:** Richi Sakaguchi, Masaki Tanaka.

**Writing – review & editing:** Richi Sakaguchi, Masaki Tanaka.

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
