## [Decision Letter · Decision Letter 0]

21 Nov 2022

PONE-D-22-19171Phenotypic screening using waveform analysis of calcium dynamics in primary cortical culturesPLOS ONE

Dear Dr. Sakaguchi,

Thank you for submitting your manuscript to PLOS ONE. After careful consideration, we feel that it has merit but does not fully meet PLOS ONE’s publication criteria as it currently stands. Therefore, we invite you to submit a revised version of the manuscript that addresses the points raised during the review process.Please ensure that your decision is justified on PLOS ONE’s publication criteria and not, for example, on novelty or perceived impact.

For Lab, Study and Registered Report Protocols: These article types are not expected to include results but may include pilot data.  Please submit your revised manuscript by Jan 05 2023 11:59PM. If you will need more time than this to complete your revisions, please reply to this message or contact the journal office at plosone@plos.org. Please include the following items when submitting your revised manuscript:A rebuttal letter that responds to each point raised by the academic editor and reviewer(s). You should upload this letter as a separate file labeled 'Response to Reviewers'.A marked-up copy of your manuscript that highlights changes made to the original version. You should upload this as a separate file labeled 'Revised Manuscript with Track Changes'.An unmarked version of your revised paper without tracked changes. You should upload this as a separate file labeled 'Manuscript'.

We look forward to receiving your revised manuscript.

Kind regards,

Ming Tatt Lee, Ph.D.

Academic Editor

PLOS ONE

https://journals.plos.org/plosone/s/fileid=ba62/PLOSOne_formatting_sample_title_authors_affiliations.pdf.

Reviewers' comments:

Reviewer's Responses to Questions

**Comments to the Author**

1. Is the manuscript technically sound, and do the data support the conclusions?

Reviewer #1: Partly

Reviewer #2: Partly

2. Has the statistical analysis been performed appropriately and rigorously? 

Reviewer #1: No

Reviewer #2: Yes

3. Have the authors made all data underlying the findings in their manuscript fully available?

Reviewer #1: No

Reviewer #2: Yes

4. Is the manuscript presented in an intelligible fashion and written in standard English?

Reviewer #1: Yes

Reviewer #2: Yes

5. Review Comments to the Author

Reviewer #1: Review Sakaguchi et al. 2022

In the present study Sakaguchi and colleagues present an in vitro screening assay based on quantitative analysis of waveforms of spontaneous calcium signals in primary cortical neurons to characterize the anti-epileptic potential of drugs. While MEA-based approaches in this direction are common (yet not cited enough in the study, calcium based approaches are underrepresented despite its advantageous technical simplicity and decreased analytical complexity.

Despite the conceptual idea and the advantage of this technically and analytically simple study is interesting many minor and major points need to be addressed to allow this study to provide solid advance for the field and co-workers in the field. Some of which are listed below:

Introduction

I don’t get the intention of the first sentence: The CNS has critical roles in homeostatic regulation. ? What does this mean, critical role in homeostatic regulation of body functions ? or did you want to emphasize that neuronal activity is homeostatically regulated, please clarify.

The introduction should also introduce and include References for MEA based screening approaches, e.g. https://doi.org/10.1038/s41598-018-28835-7 . Also at least some review references and a bit more information on epilepsy- I don’t find it sufficient to state that “ epilepsy is a type of neurological disorder”- it is the most common neurological order with still a high amount of therapy-resistant cases… I also doubt that experts would agree that lowering magnesium in slices causes “epileptic neuronal activity” (correct term is e.g. epileptiform activity).

AED not introduced

I would also not agree with the statement “ that calcium oscillations mimic epileptic activity”

Methods:

0.1 % DMSO is used as control conditions but it remains unclear if all drugs tested are also dissolve in DMSO and the final DMSO concentration is similar under all conditions.

Imaging frequency remains unclear

If I understood the given n and replic numbers correctly each experiment was performed only once with 6 replicas on the same plate. From my point of view at least 3 independent experiments with cells from at least 3 preparations should be performed, especially since spontaneous activity largely varies between cultures and thus likely also will display a variance in drug response (see also Wagenaar et al. 2006 and others).

Results

Primary cortical cultures display correlated bursting activity already under control conditions (Wagenaar et al 2006, Sun et al, 2010 etc) therefore it would be important to understand how calcium oscillations under 0-Mg conditions differ from spontaneous activity under control conditions and how much this really can be used as a model to mimic epileptiform actiform activity.

As for the negative control (normal magenesium ) I would like to have seen one clear positive control in the experimental setup, i.e. a drug that is non with a well described anti-epileptic effect in vitro, especially since the concentrations of drugs used in the study are based on therapeutic range in plasma with thus an unknown

Also a ttx control would be helpful, to understand what is the action-potential component of the observed calcium deflection.

Quality of figures is very pour, higher resolution and contrast needed, e.g. Fig 3 b cannot be judged because labels are not readable.

Figure 5a – no labels for different colour- coded symbols

Discussion

Again the first sentence is confusion- what do you mean with it is known that “intracellular calcium oscillations partially reflect neuronal activity” ?

Overall the discussion is missing on the discussion of concentration dependent effects, the advantage and disadvantage of using this reduced feature analysis based on calcium versus other e.g. MEA based analysis, the physiological correlates of analysed features and the power of the PCA based analysis .

Reviewer #2: The manuscript entitled “Phenotypic screening using waveform analysis of calcium 4 dynamics in primary cortical cultures” by Sakaguchi et al. addresses an intriguing neurophysiological approach to characterize pharmacological effects of compounds effectively and in a semi-automatic manner. I note, that the title appears a little misleading, because the reader might expect classification of neuronal phenotypes based on the Ca-waveform signatures they produce under network inputs. The manuscript is clearly written and the motivation for the study is well justified, however, I find it limited in scope and it is not clear how well the findings can be generalized to other systems. I suggest the inclusion of additional experimental data as well as the broadening of the Discussion.

Specific points.

I would suggest that the authors first demonstrate and analyze of electrical activity and Ca-dynamics of cortical cultures in control conditions without pharmacological manipulations. The ‘baseline’ activity and the temporal evolution of burst waveforms and Ca-waves would be a welcome addition to the manuscript. Indeed, synchronous activity of cortical neuronal cultures is well established, but the burst waveforms have been shown to be more variable across wells and during the maturation of those. The waveforms shown by the authors appear very homogeneous, lacking any fine structure. However, mature networks of cultured cortical neurons tend to produce prolonged bursts, often lasting tens of seconds, that exhibit interesting internal fine structure (i.e. bursts within the main burst episode). The authors demonstrate stereotypic and regular Ca-waves that might be associated with regular and accurately replayed bursts of action potentials in such networks. Did the authors select cell cultures that produce such reliable and homogeneous Ca-waves or all their networks in the 96-well plates exhibited similar activities? I would recommend to show the natural well-to-well variability of Ca-dynamics. While Fig. 1 shows examples of such Ca-waveform traces, I would suggest to display them in a less condensed format so the reader can better estimate the well-to-well variability. Also, descriptive statistics of the Ca-wave features in normal conditions would be useful to add to the manuscript.

Additionally, it is not clear whether the optical signal represents activities of multiple synchronized neurons or just single ones. If I understand correctly, network oscillations of 60 cultures in the 96-well plate are shown in Fig. 1. This should be clarified. It is hard to see the individual traces, but it appears that considerable well-to-well variability exists, and the fine structure of Ca-waves might be very apparent in some of these recordings. If that is the case, the features selected by the authors might be limited in describing the fine structure of those.

Another question is why the authors did not use interevent intervals and their variability as features. AUC is correlated with the frequency of bursts, but the interevent interval is a more informative feature. Peak number is also less informative in this regard. Clearly, drugs can boost or reduce the generation of network bursts that will manifest as clear shifts in the interevent interval distributions. Means, C.V. and other statistical descriptors can be considered for such analysis.

If I understand correctly, all the recordings were made using low-Mg external solution. At the same time, neuronal cultures, in fact, exhibit a wide repertoire of burst oscillations in normal solutions, too. Cortical neurons can certainly exhibit clear synchronized bursting at normal Mg-concentrations. Does the lowering of Mg facilitate the regularization of burs oscillations in the cultures? Are these stereotypic Ca-peaks observed only in low Mg solution? I suggest that the authors elaborate on the temporal features of Ca-waveforms/peaks in these two different scenarios.

I think the features the authors selected cannot accurately describe the peaks when they overlap in time. This can happen when two bursts occur within seconds or during prolonged superburst episodes. The authors should address this potential limitation of their technique.

It would be good to include a discussion on the generality and applicability of the author’s technique. Are these features work with other types of primary neuronal cultures or perhaps human iPSC-derived neurons?

Minor.

Abbreviations are not always explained (AED, ALS).

The motivation for the study should be better explained in the Introduction.

Line 141. How different the epileptiform burst they observe in low Mg solution are from normal activity in such cell cultures?

Line 153. Are the representative traces from individual neurons or a broader area (ROI)? Does the temporal structure (e.g. peak width) depend on the size of the ROI?

6. PLOS authors have the option to publish the peer review history of their article (what does this mean?). If published, this will include your full peer review and any attached files.

Reviewer #1: No

Reviewer #2: No

---

## [Author Response · Author response to Decision Letter 0]

23 Jan 2023

https://journals.plos.org/plosone/s/fileid=ba62/PLOSOne_formatting_sample_title_authors_affiliations.pdf.

We have ensured that this manuscript conforms to the publishing style of PLOS One.

We have uploaded our data to Dryad Digital Repository. When our manuscript is accepted, we will publish our data and specify accession number.

Reviewers' comments:

Reviewer's Responses to Questions

Comments to the Author

1. Is the manuscript technically sound, and do the data support the conclusions?

Reviewer #1: Partly

Reviewer #2: Partly

Thanks for the careful reading. In the empirical research part, we added the explanation and statistical information for selecting the experimental condition (Fig 1B and C). We added the overlaid traces of peaks of calcium oscillations in order to show that the well-to-well viability after adding compounds is small (Fig 2B). And, we conducted and showed other two replicates to represent sample-to-sample variability (S2 and S3 Figs). The information supports our conclusion profoundly. 

2. Has the statistical analysis been performed appropriately and rigorously? 

Reviewer #1: No

Reviewer #2: Yes

Thanks for the careful reading. We conducted rigorous statistical analysis in ensuring empirical research (Fig 1C) for explaining the reason why we selected this condition. And, we conducted and showed other two replicates (S1 and S2 Figs).

3. Have the authors made all data underlying the findings in their manuscript fully available?

Reviewer #1: No

Reviewer #2: Yes

Thanks for the careful reading. Our data have been uploaded to Dryad Digital Repository. When our manuscript is accepted, we will publish our data and specify accession number.

4. Is the manuscript presented in an intelligible fashion and written in standard English?

Reviewer #1: Yes

Reviewer #2: Yes

Thanks for reading our manuscript attentively.

5. Review Comments to the Author

Please use the space provided to explain your answers to the questions above. You may also include additional comments for the author, including concerns about dual publication, research ethics, or publication ethics. (Please upload your review as an attachment if it exceedS30,000 characters)

Reviewer #1: Review Sakaguchi et al. 2022

In the present study Sakaguchi and colleagues present an in vitro screening assay based on quantitative analysis of waveforms of spontaneous calcium signals in primary cortical neurons to characterize the anti-epileptic potential of drugs. While MEA-based approaches in this direction are common (yet not cited enough in the study, calcium based approaches are underrepresented despite its advantageous technical simplicity and decreased analytical complexity.

Despite the conceptual idea and the advantage of this technically and analytically simple study is interesting many minor and major points need to be addressed to allow this study to provide solid advance for the field and co-workers in the field. Some of which are listed below:

Introduction

I don’t get the intention of the first sentence: The CNS has critical roles in homeostatic regulation. ? What does this mean, critical role in homeostatic regulation of body functions ? or did you want to emphasize that neuronal activity is homeostatically regulated, please clarify.

We thank this reviewer for pointing the ambiguous of our sentence. We emphasized that CNS has critical roles in homeostatic regulation of body function as this reviewer mentioned. Thus, CNS is one of the most active area for drug discovery. We added some sentences in our introduction (L44-47).

The introduction should also introduce and include References for MEA based screening approaches, e.g. https://doi.org/10.1038/s41598-018-28835-7 . Also at least some review references and a bit more information on epilepsy- I don’t find it sufficient to state that “ epilepsy is a type of neurological disorder”- it is the most common neurological order with still a high amount of therapy-resistant cases…

We added more information and references about epilepsy and referred to existing drug-resistant patients and significance of new drugs which have novel MoAs (L63-69).

 I also doubt that experts would agree that lowering magnesium in slices causes “epileptic neuronal activity” (correct term is e.g. epileptiform activity).

AED not introduced

I would also not agree with the statement “ that calcium oscillations mimic epileptic activity”

According to the reviewers’ suggestions, we removed the expression that low-Mg condition mimicked the epileptiform activity and added the text to emphasize that our methods have been developed for analysis of neuronal firings because epileptic seizures are accompanied by increasing synchronized burst firings (L168-173).

Methods:

0.1 % DMSO is used as control conditions but it remains unclear if all drugs tested are also dissolve in DMSO and the final DMSO concentration is similar under all conditions.

We added the explanation of DMSO concentrations to our methods section (L107-108).

Imaging frequency remains unclear

If I understood the given n and replic numbers correctly each experiment was performed only once with 6 replicas on the same plate. From my point of view at least 3 independent experiments with cells from at least 3 preparations should be performed, especially since spontaneous activity largely varies between cultures and thus likely also will display a variance in drug response (see also Wagenaar et al. 2006 and others).

We agreed that variability of features and drug responses among sampling dates existed. To ensure the effectiveness of our method, we attached the supporting figures about other two replicates (S1 and S2 Figs both containing PCA plots, variance explained, and quantitative variables) and add the sentence that referred to this point (L241-242, and L314-317). 

Results

Primary cortical cultures display correlated bursting activity already under control conditions (Wagenaar et al 2006, Sun et al, 2010 etc) therefore it would be important to understand how calcium oscillations under 0-Mg conditions differ from spontaneous activity under control conditions and how much this really can be used as a model to mimic epileptiform actiform activity.

We added representative traces in 0.1 mM and 1 mM Mg2+ conditions and comparison of the number of peaks (Fig 1B and C). Other two replicates had similar tendency (data not shown). And we removed the sentences that low-Mg condition mimicked epileptiform activity.

As for the negative control (normal magenesium ) I would like to have seen one clear positive control in the experimental setup, i.e. a drug that is non with a well described anti-epileptic effect in vitro, especially since the concentrations of drugs used in the study are based on therapeutic range in plasma with thus an unknown.

According to the reviewers’ suggestions, we removed the expression that low-Mg condition mimicked the epileptiform activity, as we described above.

Also a ttx control would be helpful, to understand what is the action-potential component of the observed calcium deflection.

As advised, we applied TTX to our calcium oscillation analysis. Calcium oscillations were disappeared in 30 nM or higher concentration of TTX. In previous report, 100 nM TTX totally abolished spontaneous discharges in patch-clamp and MEA (Kitamura et al., 2005 and Halliwell et al., 2021). Our assay was comparable to these systems. The targets of TTX are mainly Nav1.1, 1.2, 1.3, 1.4 and 1.7. Then, as these systems, influx of Na+ through these voltage-gated sodium channels is essential for observed calcium oscillations. Although the main targets of Lamotrigine and lacosamide, which we used in this paper, are voltage-gated sodium channels, IC50 of lamotrigine (IC50 values are >30 uM for Nav1.1, 1.2, 1.3 and 1.6) and lacosamide (IC50 values are 50 uM for Nav 1.5) are much higher than TTX (IC50 values are <30 nM for Nav1.1, 1.2, 1.3, 1.4, and 1.7) (Qiao et al., 2015, Wang et al., 2014 and Bagal et al., 2015). Moreover, lamotrigine blocks voltage-gated calcium channels (Dibué-Adjei et al., 2017) and lacosamide selectively binds to the slow-inactivated state of the channel (Errington et al., 2006, Sheets et al., 2008, and Niespodziany et al., 2013). Thus, lamotrigine and lacosamide are different from TTX in our method. 

Quality of figures is very pour, higher resolution and contrast needed, e.g. Fig 3 b cannot be judged because labels are not readable.

Figure 5a – no labels for different colour- coded symbols

Thank you for careful reading. We amended and added the labels (Figs 3b, 5a and 5b).

Discussion

Again the first sentence is confusion- what do you mean with it is known that “intracellular calcium oscillations partially reflect neuronal activity” ?

In the previous report, calcium spiking is driven by bursts of action potentials (Shen et al., 1996). We amended the ambiguous expression to the defined expression.

Overall the discussion is missing on the discussion of concentration dependent effects, the advantage and disadvantage of using this reduced feature analysis based on calcium versus other e.g. MEA based analysis, the physiological correlates of analysed features and the power of the PCA based analysis .

We appreciate the enthusiasm on our manuscript. In the revised manuscript, we discussed that achievement of classifying compounds only when we selected optimal concentration s of compounds (L317-321), the significance of using PCA analysis (L311-317), and advantages and disadvantages of our method compared with MEA analysis (L346-356).

Reviewer #2: The manuscript entitled “Phenotypic screening using waveform analysis of calcium dynamics in primary cortical cultures” by Sakaguchi et al. addresses an intriguing neurophysiological approach to characterize pharmacological effects of compounds effectively and in a semi-automatic manner. I note, that the title appears a little misleading, because the reader might expect classification of neuronal phenotypes based on the Ca-waveform signatures they produce under network inputs. 

We amended the title from “calcium dynamics” to “synchronized calcium oscillations” in order to emphasize that we measured calcium signals not in single neuron but in a well. Additionally, we attached a movie file of measurement of calcium oscillations for clarifying calcium oscillations we captured in our method (S1 movie). 

The manuscript is clearly written and the motivation for the study is well justified, however, I find it limited in scope and it is not clear how well the findings can be generalized to other systems. I suggest the inclusion of additional experimental data as well as the broadening of the Discussion.

As advised, we added discussion about advantages and limitations compared with MEA and described the relation between electrical activity and Ca dynamics below. 

Specific points.

I would suggest that the authors first demonstrate and analyze of electrical activity and Ca-dynamics of cortical cultures in control conditions without pharmacological manipulations. 

We have tried the simultaneous measurement of electrical activity and Ca oscillation by using Fluovolt and Cal-590. We determined that we measured only calcium signals because the both signals were totally synchronized. If we can conduct the measurement of calcium oscillations on MEA plates, we could detect detailed insights of the relationship of electrical activity. However, at present, it is difficult because conditions like cell density and coating methods are different between for MEA and calcium oscillations in our lab.

The ‘baseline’ activity and the temporal evolution of burst waveforms and Ca-waves would be a welcome addition to the manuscript. Indeed, synchronous activity of cortical neuronal cultures is well established, but the burst waveforms have been shown to be more variable across wells and during the maturation of those. 

We decided day in vitro (DIV) 14 in reference to the previous report (Pacico et al., 2014). In this paper, authors reported that the calcium oscillation pattern reached plateau after DIV13. Also, we reproduced in our laboratory. We added the text for explanation (L164-165).

The waveforms shown by the authors appear very homogeneous, lacking any fine structure. However, mature networks of cultured cortical neurons tend to produce prolonged bursts, often lasting tens of seconds, that exhibit interesting internal fine structure (i.e. bursts within the main burst episode). The authors demonstrate stereotypic and regular Ca-waves that might be associated with regular and accurately replayed bursts of action potentials in such networks. Did the authors select cell cultures that produce such reliable and homogeneous Ca-waves or all their networks in the 96-well plates exhibited similar activities? I would recommend to show the natural well-to-well variability of Ca-dynamics. While Fig. 1 shows examples of such Ca-waveform traces, I would suggest to display them in a less condensed format so the reader can better estimate the well-to-well variability. Also, descriptive statistics of the Ca-wave features in normal conditions would be useful to add to the manuscript.

Also, we have tried to measure at DIV21 and 28 in 0.1 mM Mg condition. We observed C.V. of the number of peaks tended to be higher than DIV14. In order to reduce well-to-well variability, we have conducted other experiments at DIV14.

As suggested, we added the explanation for the conditioning of buffer (L168-173). To demonstrate the difference, we analyzed the number of peaks in low-Mg (0.1 mM) and normal Mg (1 mM) conditions and compared statistically (Fig 1B and C). Moreover, in order to describe that well-to-well variability after adding compounds is small, we showed the overlaid traces of peaks of calcium oscillations (Fig 2B). A trace represented averaged traces in each well.

Additionally, it is not clear whether the optical signal represents activities of multiple synchronized neurons or just single ones. If I understand correctly, network oscillations of 60 cultures in the 96-well plate are shown in Fig. 1. This should be clarified. 

We have confirmed signal fluctuations in each whole well. We amended the text to clarify that we observed calcium oscillations synchronized in each well (L166). Also, as described above, we attached a movie file of measurement of calcium oscillations for clarifying our assay (S1 movie).

It is hard to see the individual traces, but it appears that considerable well-to-well variability exists, and the fine structure of Ca-waves might be very apparent in some of these recordings. If that is the case, the features selected by the authors might be limited in describing the fine structure of those.

Another question is why the authors did not use interevent intervals and their variability as features. AUC is correlated with the frequency of bursts, but the interevent interval is a more informative feature. Peak number is also less informative in this regard. Clearly, drugs can boost or reduce the generation of network bursts that will manifest as clear shifts in the interevent interval distributions. Means, C.V. and other statistical descriptors can be considered for such analysis.

As shown in Fig 2B, waveforms of oscillations are different from each compound. We could express characters of compounds as features of waveforms using our method. It is certainly true that we could not describe all fine structures of traces. In this paper, six features we selected were sufficient because we focused on classifying positive allosteric modulators of inhibitory inputs (diazepam and clonazepam) and inhibitors of excitatory inputs (perampanel, carbamazepine, lamotrigine, and lacosamide). In contrast, we could not classify inhibitors clearly (Figs 3 and 5). Increasing other features like interevent intervals will help to express various waveforms and to characterize more compounds and more clearly. We added the outlook regarding to this point to our discussion (L314-317 and L352-356). 

If I understand correctly, all the recordings were made using low-Mg external solution. At the same time, neuronal cultures, in fact, exhibit a wide repertoire of burst oscillations in normal solutions, too. Cortical neurons can certainly exhibit clear synchronized bursting at normal Mg-concentrations. Does the lowering of Mg facilitate the regularization of burs oscillations in the cultures? Are these stereotypic Ca-peaks observed only in low Mg solution? I suggest that the authors elaborate on the temporal features of Ca-waveforms/peaks in these two different scenarios.

As we described above, control of magnesium concentration achieved the measurement of more peaks of neuronal activity in order to increase data points. To demonstrate this point, we performed a statistical analysis and added a figure (L168-173, Fig 1C). As with DIV21 and 28, C.V. of the number of peaks tended to be lower in 0.1 mM Mg condition than in 1 mM. 

I think the features the authors selected cannot accurately describe the peaks when they overlap in time. This can happen when two bursts occur within seconds or during prolonged superburst episodes. The authors should address this potential limitation of their technique.

As advised, we discussed the limitation of our method in the discussion section (L346-356)

It would be good to include a discussion on the generality and applicability of the author’s technique. Are these features work with other types of primary neuronal cultures or perhaps human iPSC-derived neurons?

Thank you for the suggestion. Calcium oscillations in cardiomyocytes and iPS-derived neurons were observed and applied for gaining pharmacological insights and discovery of drug targets (Psaras et al., 2021, Huang et al., 2021). We added the description and references (L358-360).

Minor.

Abbreviations are not always explained (AED, ALS).

Thank you for careful reading. We defined abbreviations upon first appearance in the text.

The motivation for the study should be better explained in the Introduction.

We have developed this method for deconvolution of MoAs novel drug candidates have. We added this motivation in Introduction (L55-62).

Line 141. How different the epileptiform burst they observe in low Mg solution are from normal activity in such cell cultures?

In low Mg condition, the number of peaks increases (Fig 1B and C). However, as suggested by Reviewer#1, it is not general to regard this pattern as epileptiform activity, then we removed this description.

Line 153. Are the representative traces from individual neurons or a broader area (ROI)? Does the temporal structure (e.g. peak width) depend on the size of the ROI?

As described above, we observed bursts of network in each whole well. We amended the text to clarify this point (L166).

6. PLOS authors have the option to publish the peer review history of their article (what does this mean?). If published, this will include your full peer review and any attached files.

Do you want your identity to be public for this peer review? For information about this choice, including consent withdrawal, please see our Privacy Policy.

Reviewer #1: No

Reviewer #2: No 

We agreed to publish the peer review history of this article alongside the article.

---

## [Decision Letter · Decision Letter 1]

20 Feb 2023

PONE-D-22-19171R1Phenotypic screening using waveform analysis of synchronized calcium oscillations in primary cortical culturesPLOS ONE

Dear Dr. Sakaguchi,

Thank you for submitting your manuscript to PLOS ONE. After careful consideration, we feel that it has merit but does not fully meet PLOS ONE’s publication criteria as it currently stands. Therefore, we invite you to submit a revised version of the manuscript that addresses the points raised during the review process.

We look forward to receiving your revised manuscript.

Kind regards,

Ming Tatt Lee, Ph.D.

Academic Editor

PLOS ONE

Journal Requirements:

Additional Editor Comments (if provided):

Kindly address the comments by the reviewers.

Reviewers' comments:

Reviewer's Responses to Questions

**Comments to the Author**

1. If the authors have adequately addressed your comments raised in a previous round of review and you feel that this manuscript is now acceptable for publication, you may indicate that here to bypass the “Comments to the Author” section, enter your conflict of interest statement in the “Confidential to Editor” section, and submit your "Accept" recommendation.

Reviewer #1: (No Response)

Reviewer #2: All comments have been addressed

2. Is the manuscript technically sound, and do the data support the conclusions?

Reviewer #1: Partly

Reviewer #2: (No Response)

3. Has the statistical analysis been performed appropriately and rigorously? 

Reviewer #1: Yes

Reviewer #2: (No Response)

4. Have the authors made all data underlying the findings in their manuscript fully available?

Reviewer #1: Yes

Reviewer #2: (No Response)

5. Is the manuscript presented in an intelligible fashion and written in standard English?

Reviewer #1: Yes

Reviewer #2: (No Response)

6. Review Comments to the Author

Reviewer #1: Sakaguchi and colleagues have significantly improved the manuscipt. Yet, two more points remain to be adressed:

Imaging frequency is still unclear and should be addede to Material and Methods.

I would still highly recommend to include the “Other two replicates” into the dataset and not just state (data not shown). To base scientific conclusions or validation of methodology on a single biological replica is not sufficient from my point of view. Especially if individual and “batch-to-batch” (i.e. culture preparation-to-culture preparation) are well described. At least this limitation of the dataset should be mentioned in the manuscript.

Reviewer #2: The authors responded adequately to my questions and suggestions. I recommend the acceptance of the manuscript for publication.

7. PLOS authors have the option to publish the peer review history of their article (what does this mean?). If published, this will include your full peer review and any attached files.

Reviewer #1: No

Reviewer #2: No

---

## [Author Response · Author response to Decision Letter 1]

1 Mar 2023

Journal Requirements:

We thank for indicating our reference style was wrong and incomplete. We amended the style to the Vancouver style. Also, we omitted an un-quoted reference and checked the references again for any omissions.

1. If the authors have adequately addressed your comments raised in a previous round of review and you feel that this manuscript is now acceptable for publication, you may indicate that here to bypass the “Comments to the Author” section, enter your conflict of interest statement in the “Confidential to Editor” section, and submit your "Accept" recommendation.

Reviewer #1: (No Response)

Reviewer #2: All comments have been addressed 

Thanks for the careful reading.

2. Is the manuscript technically sound, and do the data support the conclusions?

Reviewer #1: Partly

Reviewer #2: (No Response)

Thanks for the careful reading. We added all replicates to our datasets (Figs 3, 5, S1 and S2). Also, we amended the text to mention the variabilities of drug responses caused by sampling dates (L247-250, L275, L316, and L351-355).

3. Has the statistical analysis been performed appropriately and rigorously? 

Reviewer #1: Yes

Reviewer #2: (No Response)

We thank all the reviewers for careful and constructive reviews. 

4. Have the authors made all data underlying the findings in their manuscript fully available?

Reviewer #1: Yes

Reviewer #2: (No Response) 

Thank you for favorable responses.

5. Is the manuscript presented in an intelligible fashion and written in standard English?

Reviewer #1: Yes

Reviewer #2: (No Response)

Thanks for reading our manuscript attentively.

6. Review Comments to the Author

Reviewer #1: Sakaguchi and colleagues have significantly improved the manuscipt. Yet, two more points remain to be adressed:

Imaging frequency is still unclear and should be addede to Material and Methods.

I apologized for missing this comment. As advised, we added imaging frequency (27 Hz) to the Materials and methods section (L146).

I would still highly recommend to include the “Other two replicates” into the dataset and not just state (data not shown). To base scientific conclusions or validation of methodology on a single biological replica is not sufficient from my point of view. Especially if individual and “batch-to-batch” (i.e. culture preparation-to-culture preparation) are well described. At least this limitation of the dataset should be mentioned in the manuscript.

We appreciate the enthusiasm on our manuscript. We totally agreed with this reviewer. To comprehend the robustness of our method in spite of variability of drug responses derived from conditions of primary culture, we showed PCA plots, the relative contributions of the principal components and contribution of the different features of all three replicates (Figs 3, 5, S1 and S2). Also, we amend the text to reflect the results of all replicates (L247-250, and L275). Moreover, we declared that there were differences of drug responses depending on sampling dates and we focused on segregation of compounds in this paper because all three replicates prepared individually (L316, and L351-355).

Reviewer #2: The authors responded adequately to my questions and suggestions. I recommend the acceptance of the manuscript for publication.

We appreciate favorable comments on our revision.

7. PLOS authors have the option to publish the peer review history of their article (what does this mean?). If published, this will include your full peer review and any attached files.

Do you want your identity to be public for this peer review? For information about this choice, including consent withdrawal, please see our Privacy Policy.

Reviewer #1: No

Reviewer #2: No 

We agreed to publish the peer review history of this article alongside the article.

---

## [Decision Letter · Decision Letter 2]

13 Mar 2023

Phenotypic screening using waveform analysis of synchronized calcium oscillations in primary cortical cultures

PONE-D-22-19171R2

Dear Dr. Sakaguchi,

We’re pleased to inform you that your manuscript has been judged scientifically suitable for publication and will be formally accepted for publication once it meets all outstanding technical requirements.

Kind regards,

Ming Tatt Lee, Ph.D.

Academic Editor

PLOS ONE

Additional Editor Comments (optional):

Reviewers' comments:

Reviewer's Responses to Questions

**Comments to the Author**

1. If the authors have adequately addressed your comments raised in a previous round of review and you feel that this manuscript is now acceptable for publication, you may indicate that here to bypass the “Comments to the Author” section, enter your conflict of interest statement in the “Confidential to Editor” section, and submit your "Accept" recommendation.

Reviewer #1: All comments have been addressed

2. Is the manuscript technically sound, and do the data support the conclusions?

Reviewer #1: Yes

3. Has the statistical analysis been performed appropriately and rigorously? 

Reviewer #1: Yes

4. Have the authors made all data underlying the findings in their manuscript fully available?

Reviewer #1: Yes

5. Is the manuscript presented in an intelligible fashion and written in standard English?

Reviewer #1: Yes

6. Review Comments to the Author

Reviewer #1: (No Response)

7. PLOS authors have the option to publish the peer review history of their article (what does this mean?). If published, this will include your full peer review and any attached files.

Reviewer #1: **Yes: **Anne Sinning

<quillbot-extension-portal></quillbot-extension-portal>

---

## [Editor Report · Acceptance letter]

20 Apr 2023

PONE-D-22-19171R2 

Phenotypic screening using waveform analysis of synchronized calcium oscillations in primary cortical cultures 

Dear Dr. Sakaguchi:

I'm pleased to inform you that your manuscript has been deemed suitable for publication in PLOS ONE. Congratulations! Your manuscript is now with our production department. 

Kind regards, 

on behalf of

Dr. Ming Tatt Lee 

Academic Editor

PLOS ONE